# Antifungal activity of volatile compounds generated by endophytic fungi *Sarocladium brachiariae* HND5 against *Fusarium oxysporum* f. sp. *cubense*

**Yang Yang**[1,2], **Yipeng Chen**[1,2], **Jimiao Cai**[1,2], **Xianbao Liu**[1,2], **Guixiu Huang**[1,2]*

**1** Environment and Plant Protection Institute, Chinese Academy of Tropical Agricultural Sciences, Haikou, China, **2** Key Laboratory of Integrated Pest Management on Tropical Grops, Ministry of Agriculture and Rural Affairs, Beijing, China

* hgxiu@catas.cn

**Data Availability Statement:** All relevant data are within the manuscript and its Supporting Information files.

## Abstract

The soil-born filamentous fungal pathogen *Fusarium oxysporum* f. sp. *cubense* (FOC), which causes vascular wilt disease in banana plants, is one of the most economically important Fusarium species. Biocontrol using endophytic microorganisms is among the most effective methods for controlling banana Fusarium wilt. In this study, volatile organic compounds (VOCs) showed strong antifungal activity against FOC. Seventeen compounds were identified from the VOCs produced by endophytic fungi *Sarocladium brachiariae* HND5, and three (2-methoxy-4-vinylphenol, 3,4-dimethoxystyrol and caryophyllene) showed antifungal activity against FOC with 50% effective concentrations of 36, 60 and 2900 µL/L headspace, respectively. Transmission electron microscopy (TEM) and double fluorescence staining revealed that 2-methoxy-4-vinylphenol and 3,4-dimethoxystyrol damaged the plasma membranes, resulting in cell death. 3,4-dimethoxystyrol also could induce expression of chitin synthases genes and altered the cell walls of FOC hyphae. Dichloro-dihydro-fluorescein diacetate staining indicated the caryophyllene induced accumulation of reactive oxygen species (ROS) in FOC hyphae. FOC secondary metabolism also responded to active VOC challenge by producing less fusaric acid and expressions of genes related to fusaric acid production were interrupted at sublethal concentrations. These findings indicate the potential of *S. brachiariae* HND5 as a biocontrol agent against FOC and the antifungal VOCs as fumigants.

## Introduction

The soil-born filamentous fungus *Fusarium oxysporum* f. sp. *cubense* (FOC) is one of the most economically important *Fusarium* species because it is responsible for vascular wilt disease in banana plants, which is the most destructive banana disease and leads to serious crop losses in banana plantations [1, 2]. This pathogen can infect banana from root and invade the xylem vessels and eventually cause a lethal wilting of the infected plants [3]. The phytotoxic

**Funding:** YY is founded by Hainan Provincial Natural Science Foundation of China [Grant 319QN268]. The funders had no role in study design, data collection and analysis, decision to publish, or preparation of the manuscript.

**Competing interests:** The authors have declared that no competing interests exist.

secondary metabolites fusaric acid is the main toxic molecule contributes to the pathogenicity of FOC during infection of banana plants [3]. Four races of FOC attack different banana cultivars. Among them, race 1 destroyed almost all Gros Michel bananas worldwide in the early 1900s. The introduction of the Fusarium-resistant Cavendish cultivar saved the banana industry until the 1960s. However, FOC race 4 is currently affecting the Cavendish cultivar [4]. Various methods have been developed to control FOC-induced wilt, including the destruction of diseased plants, sanitary measures, use of disease-free tissue culture planting materials, and use of tolerant plant varieties. Chemicals such as soil disinfectants and fumigants are also widely used to manage this disease [5]. However, the abuse of synthetic antifungal agents may pose risks to the environment and human health [6]. Therefore, it is essential to develop alternative, environmentally friendly methods to control FOC. For this purpose, biological control agents have attracted increasing attention because of their low mammalian toxicity, low target specificity, and environmental friendliness [7, 8].

Endophytic fungi, by definition, reside in the tissues beneath the epidermal cell layers and cause no apparent harm to the host [9]. As they can confer abiotic and biotic stress tolerance and increase the biomass of the host plant, many endophytic fungi have been studied as biological control agents, including species of *Sarocladium* (reallocated from *Acremonium* genus), *Aspergillus*, *Fusarium* and *Penicillium* [10, 11, 31]. Among them, *Sarocladium* species are well studied and suitable for developing biological control agents because, as fungal endophytes, they can easily colonize host plants [12]. *Sarocladium* species are known to produce a wide range of bioactive compounds; thus, they could inhibit pathogen growth directly or indirectly via the stimulation of induced systemic resistance [13–15]. Additionally, *Sarocladium* species might help host plants accumulate nutrients and increase organic nitrogenous compounds [16].

In addition to the well-documented suites of soluble antimicrobial compounds found in endophytic fungi, these species may also emit a wide range of volatile organic compounds (VOCs) with strong inhibitory activity against microbial competitors [17, 18]. VOCs are carbonbased solids and liquids that readily enter the gas phase by vaporizing at 0.01 kPa at a temperature of approximately 20°C; they include acids, alcohols, alkyl pyrones, ammonias, esters, hydrogen cyanides, ketones, and lipids [19]. Different kinds and amounts of VOCs are produced by microorganisms during both primary and secondary metabolism [20]. VOCs emitted by microorganisms have a variety of applications; for example, they are used to indicate biocontamination in the food industry and in indoor environments, and to identify and separate microorganisms [21–23]. In recent years, VOCs produced by microorganisms have been shown to be effective and eco-friendly biocontrol agents [24, 25]. Most of these VOCs have anti-bacterial or antifungal activity, and some can induce defence responses and promote plant growth [26, 27].

In a survey on the diversity of the endophytic fungi of *Brachiaria brizantha*, we isolated the strain HND5, which can effectively inhibit the growth of FOC mycelia. Based on the LSU (the large subunit rDNA), ITS (rDNA transcribed spacer region) sequences along with the culture morphology and whole genome sequence data, we characterized HND5 as a new species, *Sarocladium brachiariae* [28, 29]. Although HND5 has anti-phytopathogen activity, the active compounds and the mechanism of action remain unknown. In this study, we found that the VOCs emitted by HND5 significantly affect FOC growth. Three antifungal VOCs were identified from HND5, and the 50% effective concentration (EC50) values of these VOCs were analysed. Biochemical, microscopic and molecular biological analyses revealed that the antifungal VOCs resulted in the leakage of cytoplasm, cell death, inhibition of spore germination, differential gene expression, and significant alterations to secondary metabolism in FOC. In summary, the results of this study indicate that the VOCs emitted by HND5 show potential as biological control agents against FOC in agricultural production systems.

## Materials and methods

### Microorganisms and culturing conditions

The antagonistic strain *Sarocladium brachiariae* HND5. (China General Microbiology Culture Collection Center accession No. 2192) [30, 31] and *Fusarium oxysporum* f. sp. *cubense*, which exhibits high virulence against banana, was used in this study [32, 33]. The fungal strains used in this study were grown on potato-dextrose agar (PDA) plates at 28˚C for 3–7 d. To produce microconidia, 5 mm-discs from FOC culture plates were placed into potato dextrose broth (PDB) at 28˚C under shaking at 170 rpm/min for 3 d.

### Antagonistic assay of VOCs emitted from HND5 against FOC mycelium growth and conidia germination

The VOCs produced by HND5 were tested according to the method of Raza with some modifications [34]. A dual-plate experiment was used to evaluate the antifungal activities of the VOCs produced by HND5 on PDA medium against FOC. HND5 was incubated in petri plates containing PDA agar at 28˚C for 7 d. Subsequently, each HND5 plate was covered with a petri plate containing PDA incubated with a 5-mm-diameter plug of FOC, and the two plates were sealed with parafilm to obtain a double-plate chamber and incubated at 28˚C for 7 d. The control plates incubated with FOC plugs were covered with petri plates containing only PDA medium. All treatments were performed in triplicate. The mycelia diameters were measured every 24 h during incubation. For antagonistic assay against conidia germination, the FOC plug was replaced with 100 μL $10^2$ cfu/mL conidia, and the incubation time was reduced to 48 h. Active carbon was used as an absorbent to remove VOCs in the HND5 and FOC assay according to Gong's method with minor modification [35].

### Collection and analysis of VOCs

For the production and collection of VOCs, 5-mm-diameter discs of HND5 from PDA culture plates were incubated in vials with 50 mL of headspace (HS) containing 20 ml PDA medium. The vials were closed with autoclaved rubber plugs and parafilm and incubated at 28˚C for 7 d [36]. Three different solid-phase micro-extraction (SPME) fibres were used in this study: poly-dimethylsiloxane (PDMS), 100 μm; PDMS/divinylbenzene (DVB), 65 μm; DVB/ Carboxen (CAR)/PDMS, 50/30 μm (Supelco, Vienna, Austria). For extraction, SPME fibres penetrated into the HS vials for 30 min without agitation. The samples were analysed with an Agilent 7890A gas chromatograph coupled to an Agilent 5975 mass-selective detector (MSD; Agilent, Waldbronn, Germany). An HP-5MS column (30 m, 0.25-mm inner diameter, Agilent, Waldbronn, Germany) was used for sample separation. SPME fibres were desorbed at 250˚C for 5 min. The oven program was as follows: 60˚C (hold 2 min), 10˚C/min to 100˚C, 5˚C/min to 180˚C, 20˚C/min to 240˚C (hold 5 min). The MSD parameters were as follows: electron impact ionisation at 70 eV, source temperature of 230˚C, quadrupole temperature of 150˚C, solvent delay of 2.2 min, full scan (50–500 amu).

### Antagonistic assay of synthetic compounds against FOC growth and EC50 detection

The pure compounds of the three identified VOCs produced by HND5 were purchased from different companies: 2-methoxy-4-vinylphenol (99%) was purchased from Aladdin (Shanghai, China); and 3,4-dimethoxystyrol (technical grade) and (-)-trans-caryophyllene (98.5%) were obtained from Sigma-Aldrich (Vienna, Austria). Dual-plate experiments were used to test the antifungal activities of the compounds. petri dishes (9 cm diameter, the volume of free space

was 90 mL) containing PDA medium incubated with FOC plugs were covered with petri dishes containing sterile filter papers ($15 \times 20$ mm) containing different amounts of VOCs: 2-methoxy-4-vinylphenol: 0.5, 1, 5, 10, and 20 μL; 3,4-dimethoxystyrol: 1, 5, 10, 20, and 40 μL; and caryophyllene: 10, 20, 40, 80, and 160 μL. The control plates were covered with petri dishes containing only sterile filter papers. The plates were incubated at 28˚C for 7 d, and the mycelia diameters were measured every 24 h during incubation.

## Transmission electron microscopy (TEM) observation of FOC

PDA plates with FOC plugs were first incubated at 28˚C for 5 d, and the EC50 concentrations of different VOCs were added separately onto the covers. After 12 h, the VOC-treated and untreated mycelia were extracted and fixed with 2.5% glutaraldehyde overnight. The fixed cells were rinsed three times for 10 minutes with 100 mM phosphate buffer, postfixed for 3 h in 1% osmium tetroxide, and dehydrated using an ethanol gradient. The samples were then embedded in Epon 812, sectioned using an ultramicrotome, and examined under a Hitachi HT-7700 transmission electron microscope [37].

## LIVE/DEAD fungus viability staining

A mixed stain consisting of fluorescein diacetate (FDA) and propidium iodide (PI) was used to test cell viability. When used at a proper concentration, live fungal cells with intact cell membranes show green fluorescence, while dead fungal cells with damaged cell membranes show red fluorescence [38]. The hyphae of FOC treated with VOCs at EC50 concentration were extracted from the plates and resuspended in 10 mM PBS (phosphate buffered saline) buffer (pH 7.4) with 100 μg/ml FDA and 3 μg/ml PI. After 10 min of incubation at 25˚C in the dark, the hyphae were washed two times with 10 mM PBS buffer (pH 7.4). The samples were checked with a Nikon NI/E microscope.

## Detection of reactive oxygen species (ROS)

The probe dichloro-dihydro-fluorescein diacetate (DCFH-DA; Solarbio Science & Technology, Beijing, China) was used to detect the change in reactive oxygen species (ROS) in FOC hyphae after treatment with VOCs at sublethal concentrations (2-methoxy-4-vinylphenol, benzene (2M4V) and 3,4-dimethoxystyrol (34D): 1 μL/plate, caryophyllene (β-C): 40 μL/plate). The hyphae of VOC-treated FOC were extracted from the plates and resuspended in 10 mM PBS buffer (pH 7.4) with 10 μg/ml DCFH-DA. After incubation for 30 min at 20˚C, the hyphae were washed twice with 10 mM PBS buffer (pH 7.4). The samples were checked with a Nikon NI/E microscope.

## Extraction and detection of fusaric acid

Autoclaved corn kernels with 45% added water were used to test fusaric acid production using the method of Bacon with minor modification [39]. Corn kernels (10 g) mixed with 1 ml $10^8$ cfu/ml FOC spores were added into 9-cm-diameter plates, and VOCs at sublethal concentrations (2M4V and 34D: 1 μL/plate, β-C: 40 μL/plate) were added onto the covers. The plates were sealed with parafilm and incubated at 28˚C in the dark. After 4 weeks, 5-g samples were ground in a homogenizer to a uniform consistency in 20 ml of 1:1 methanol-$KH_2PO_4$ (pH 3.0). The ground samples were centrifuged at 12000 rpm for 20 min, and the pH of the supernatant was adjusted to 3.0 with HCl. The acidified supernatants were extracted three times with 30 ml of methylene chloride. The methylene chloride extracts were combined, and the solvent was removed under vacuum at 40˚C using a rotary evaporator. The remaining residues

were dissolved with 1 ml methanol and analysed by high-performance liquid chromatography (HPLC). A Waters e2695-2998 PDA HPLC system with an Agilent HC-C18 column ($4.6 \times 250$ mm) was employed to analyse fusaric acid. Quantification was performed as previously described [40].

### RNA extraction and quantitative real-time polymerase chain reaction (PCR) analysis

PDA plates with FOC plugs were first incubated at 28˚C for 5 d, and the sublethal concentrations (2M4V and 34D: 1 μL/plate, β-C: 40 μL/plate) of different VOCs were then added separately onto the covers. After 12 h, the VOC-treated and untreated mycelia were removed, lyophilized, and ground in liquid nitrogen. Total RNA was extracted from the mycelia using a RNAsimple Total RNA Kit [Tiangen Biotech (Beijing) Co., Ltd] according to the manufacturer's instructions. First-strand cDNA was synthesized using a FastKing Reverse Transcription Kit (Tiangen) with random hexamer primers. The resulting cDNA was used as the template for subsequent polymerase chain reaction (PCR) amplification. Quantitative real-time PCR (qRT-PCR) was performed using a Talent qPCR PreMix (SYBR Green; Tiangen) in a 7500 Fast Real-Time PCR Detection System. The actin gene was used as the internal reference for normalization. Primers for these genes (Actin, FOIG_00580, FOIG_06735, FOIG_06738, FUB2 and FUB5) are listed in S4 Table.

## Results

### VOCs of HND5 inhibit mycelium growth and conidium germination in FOC

In this study, a dual-plate assay system was used to test the antifungal activities of VOCs emitted by HND5 against FOC mycelium growth and conidium germination because there was no physical contact between FOC and HND5 colony. As shown in Fig 1, the diameter of untreated FOC mycelium reached 3 cm after three days of incubation, and untreated conidia germinated after 48 h on the PDA plate. In contrast, the HND5-treated mycelium grew to a diameter of only 1 cm, and no conidia germinated. To verify that the VOCs produced by HND5 were directly responsible for the antifungal activity, charcoal, which can absorb VOCs, was added into the system. The added charcoal reduced the antifungal activity of HND5, indicating that the activity against FOC was indeed caused by the VOCs produced by HND5 (Fig 1).

### Gas chromatography/mass spectrometry (GC/MS) analysis of VOCs produced by HND5

Because SPME fibres with different coating materials have different absorption characteristics [36], three different SPME fibre coatings (PDMS, 100 μm; DVB, 65 μm; DVB/CAR/PDMS, 50/30 μm) were used in this study to obtain a complete picture of the VOCs produced by HND5. The total ion current chromatograms and identified compounds of the HND5 cultures and pure PDA medium after HS extraction were compared (S1 Table: PDMS, 100 μm results; S2 Table: DVB, 65 μm results; S3 Table: DVB/CAR/PDMS, 50/30 μm). To exclude background signals from agar, the SPME fibre coating, and the stationary phase, only compounds that were not present in the chromatograms of blank agar medium and with intensities bigger than $1 \times 10^5$ counts per second were further considered. In total, 17 compounds were identified in the VOCs produced by HND5; these compounds belonged to alkenes, alkyls, ketones, esters and aromatic hydrocarbons (Table 1).

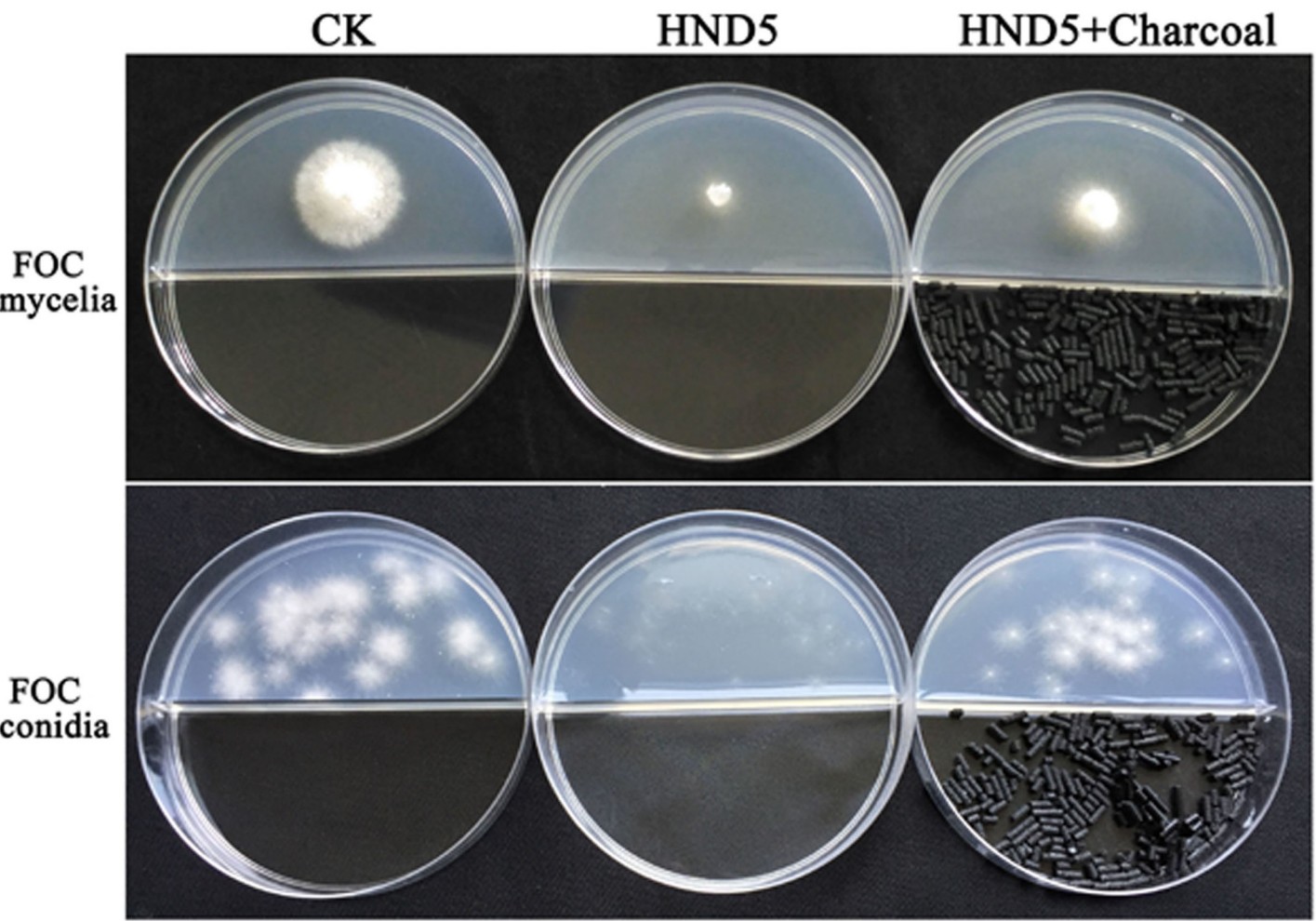

**Fig 1. VOCs of HND5 inhibit mycelia growth and conidia germination of FOC.** CK: PDA with FOC plug or FOC conidia covered with petridish containing PDA medium; HND5 treatment: PDA with FOC petri or FOC conidia covered with petridish containing PDA medium incubated with HND5 plug; HND5+Charcoal treatment: HND5 treatment with 5g charcoal. All plates were incubated at 28°C, 3 d for mycelia growth and 48 h for conidia germination.

### Antifungal activities of individual VOC against FOC and EC50 analysis

Among the 17 identified VOC compounds, only three (2-methoxy-4-vinylphenol, benzene, 2M4V; 3,4-dimethoxystyrol, 34D; and caryophyllene, β-C) were available from reagent companies. These three VOCs were selected for further testing of antifungal activity against FOC. As shown in Fig 2, all three VOCs inhibited the growth of FOC mycelia at a concentration of 10 μL/plate. 2M4V and 34D showed a stronger antifungal activity than β-C at the same concentration. We also tested a range of concentrations to determine the EC50 values of the selected VOCs against FOC. The results showed that the inhibitory activities of the VOCs increased with VOC concentration (Fig 3). Colony diameter was measured, and the EC50 was calculated via statistical analysis. As the volume of free space of the plates was 90 mL, we then transformed unit μL/plate into μL/L. The EC50 values of 2M4V, 34D and β-C against FOC were found to be 36, 60 and 2900 μL/L, respectively.

### Micro- and ultrastructural changes to FOC hyphae induced by VOCs

To determine the mechanism of VOC activity, TEM was used to evaluate the ultrastructural damage to hyphae caused by the selected VOCs. As shown in Fig 4, the normal FOC hyphae

**Table 1. VOCs produced by HND5.**

| RT[a] (min) | Compounds[b] | Releative peak area[c] (%) |
|---|---|---|
| 5.105 | 1-Decene | 4.79% |
| 6.525 | Hexacosane | 4.53% |
| 7.397 | Cyclopropane, nonyl- | 19.36% |
| 7.987 | 3-Methyl-3,5—(cyanoethyl)tetrahydro-4-thiopyranone | 1.12% |
| 8.003 | Cyclododecane | 6.55% |
| 10.474 | 3-Ethyl-2-nonanone | 1.43% |
| 11.842 | 3-Tridecene, (Z)- | 1.33% |
| 12.602 | 2-Methoxy-4-vinylphenol* | 1.86% |
| 13.184 | Cyclohexene, 4-ethenyl-4-methyl-3-(1-methylethenyl)-1-(1-methylethyl)-, (3R-trans)- | 1.47% |
| 13.833 | 3,4-Dimethoxystyrol* | 30.14% |
| 14.03 | Cyclopentanecarboxylic acid, 4-isopropylidene-2-methoxymethyl-, methyl ester | 2.27% |
| 15.202 | Caryophyllene* | 3.91% |
| 15.569 | Bicyclo[3.1.1]heptan-2-one, 6,6-dimethyl- | 1.70% |
| 15.834 | 5,9-Undecadien-2-one, 6,10-dimethyl-, (Z)- | 3.75% |
| 16.586 | Naphthalene, 1,2,4a,5,6,8a-hexahydro-4,7-dimethyl-1-(1-methylethyl)- | 6.99% |
| 17.219 | Naphthalene, 1,2,3,5,6,8a-hexahydro-4,7-dimethyl-1-(1-methylethyl)-, (1S-cis)- | 3.19% |
| 20.544 | .alpha.-Cadinol | 5.60% |

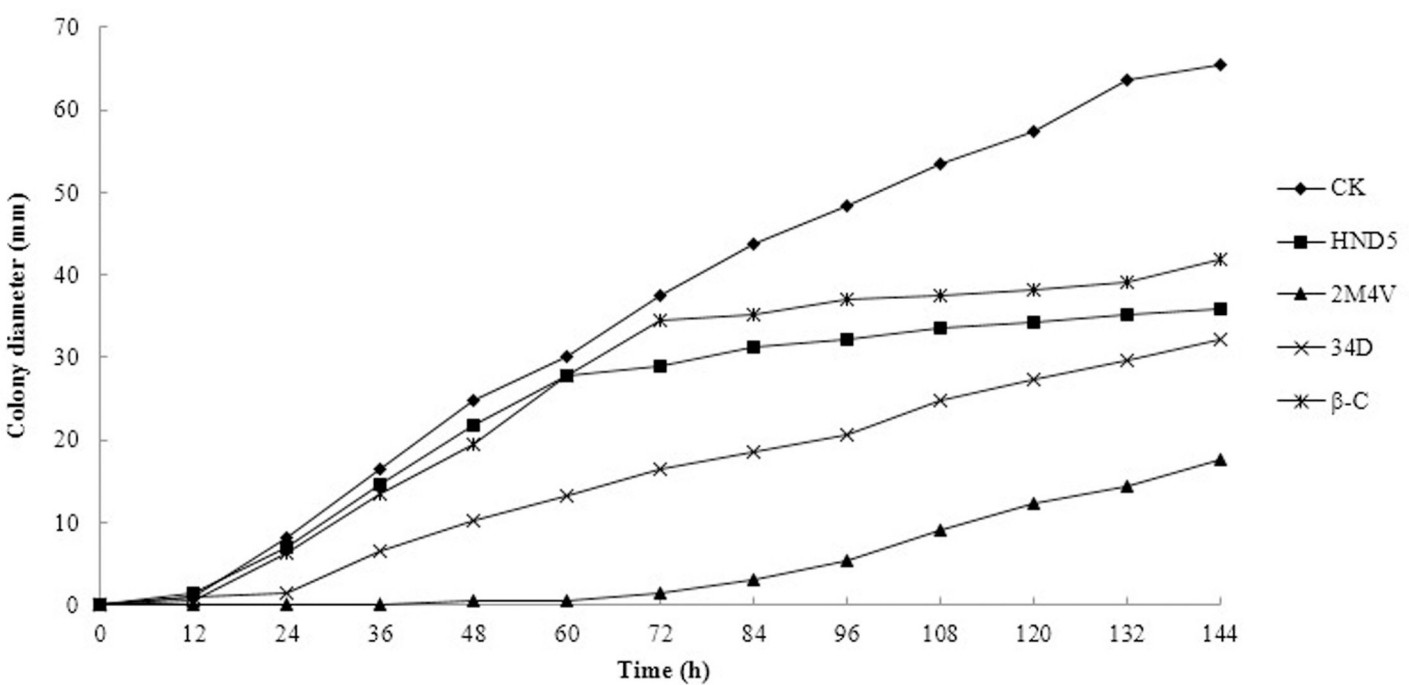

**Fig 2. The effect of selected VOCs on FOC.** 10 μL different VOCs were added separately. All plates were incubated at 28˚C.

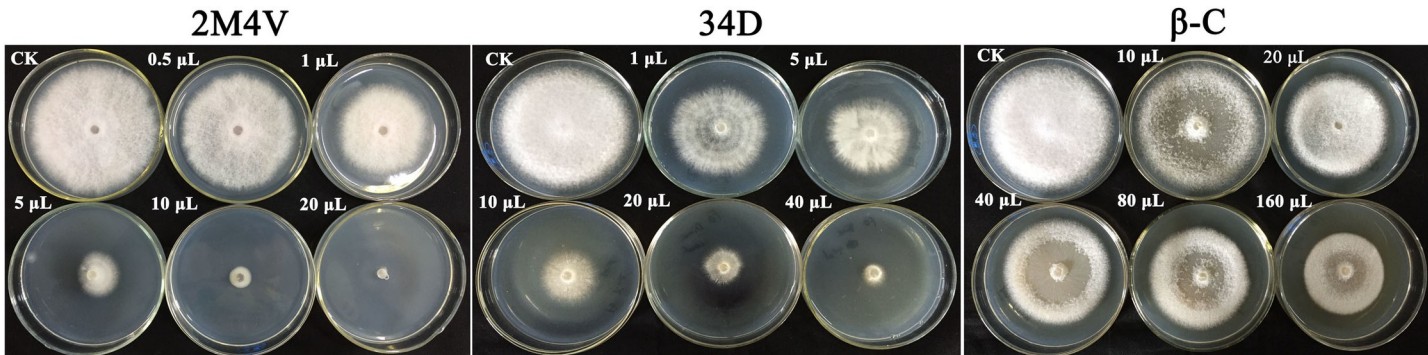

**Fig 3. EC50 analysis of selected VOCs against FOC.** All plates were incubated at 28˚C for 7 d. 2M4V: 2-Methoxy-4-vinylphenol; 34D: 3,4-Dimethoxystyrol; β-C: Caryophyllene.

showed distinct cell walls, intact plasma membranes, uniformly distributed electrodense cytoplasm and clearly visible cell organelles. In contrast, the FOC hyphae treated with 2M4V and 34D showed completely different and irregular structures without intact plasma membranes, smeared cytoplasm and leaked cell content. In addition, the cell walls of the 34D-treated hyphae were two to three times thicker than those in the other groups. Treatment with the EC50 concentration of β-C did not drastically change the hypha structure; the β-C-treated hyphae had distinct cell walls, intact plasma membranes and visible cell organelles (Fig 4).

## The selected VOCs caused cell death in FOC

Double fluorescent staining with FDA/PI was used to analyse the live/dead cells in combination with fluorescence microscopy. FDA is an enzyme activity probe that is recognized by nonspecific esterases; this recognition releases green fluorescence once it enters living cells, thus serving as an indicator of live cells. PI fluoresces red in response to membrane damage and is used as an indicator of dead cells. As shown in Fig 5, the untreated hyphae were outlined by green fluorescence (live cells), and few dead cells (red fluorescence) were observed. In contrast, after treatment with the EC50 concentration of 2M4V or 34D, the proportions of red-fluorescent hyphae cells increased, and the green fluorescence became blurry. In contrast, most β-C-treated hyphae showed green fluorescence with few dead cells, as for the untreated hyphae. Combined with the TEM results, these observations indicate that 2M4V and 34D destroyed the hyphae cell membranes, thereby inhibiting the growth of FOC, whereas β-C inhibited FOC through a different route.

## The selected VOCs affect chitin synthesis in FOC

Chitin is one of the major components of cell walls in FOC and play an important role in pathogenesis [41]. TEM analysis showed that cell walls of 34D-treated FOC hyphae were thicker than normal hyphae (Fig 4). Thus, we hypothesized that 34D could affect chitin synthesis in FOC. We analysed the expression levels of three different types of chitin synthase gene related with pathogenicity in FOC with or without VOCs treatment (at sublethal concentration) with quantitative real-time PCR: Class V (FOIG_06738) [42], ChsVb (FOIG_06735) [43] and Class 4 (FOIG_00580) [44]. Result showed expression level of ChsVb (FOIG_06735) and Class V (FOIG_06738) chitin synthase genes increased significantly after 34D treatment (Fig 6).

## VOC-induced accumulation of reactive oxygen species (ROS) in FOC

High concentrations of ROS are harmful to cells and can result in cell death [45]. To determine whether FOC cells accumulate ROS as a result of treatment with VOCs at sublethal

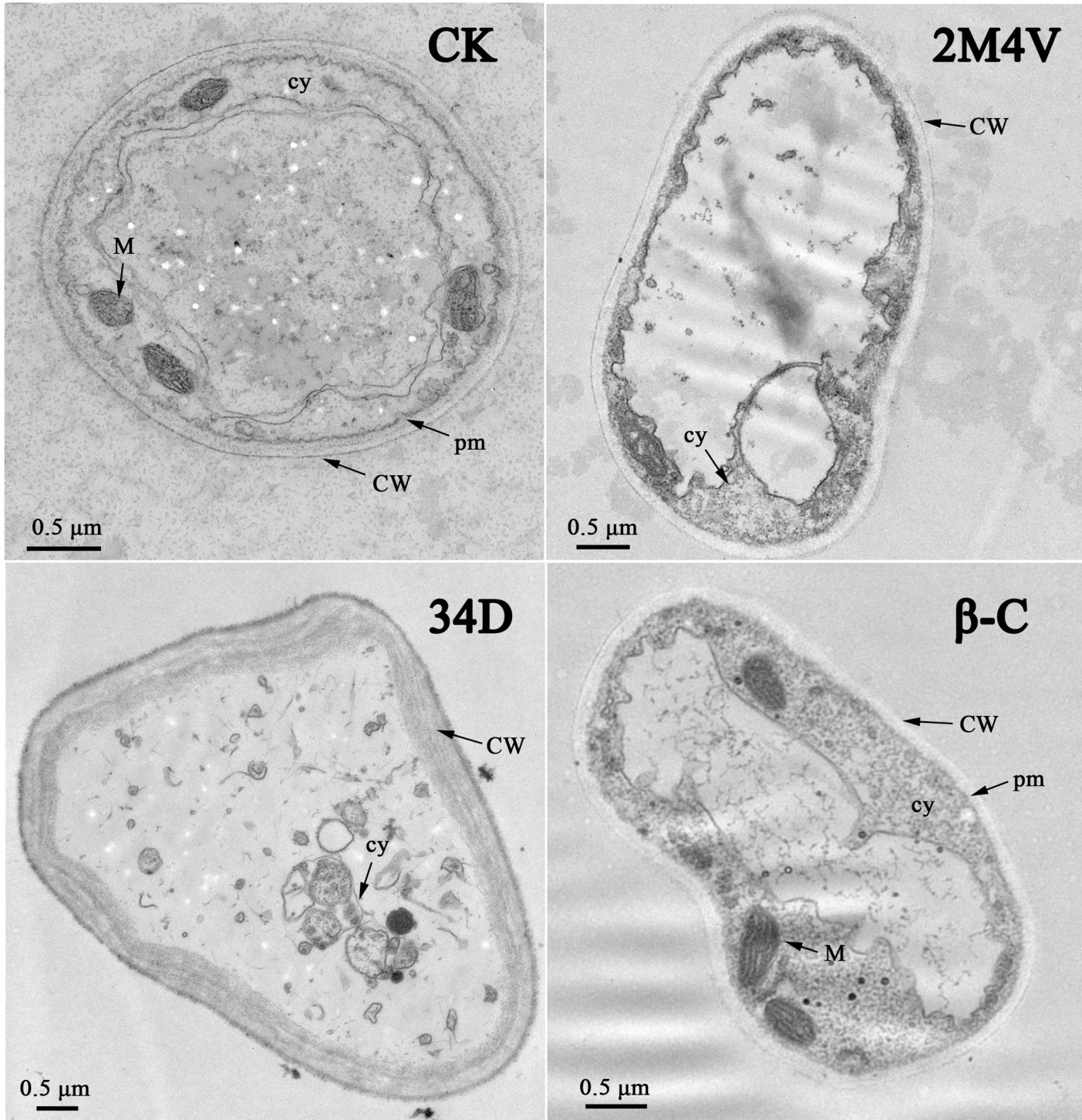

**Fig 4. Ultrastructural effects of EC50 concentration of selected VOCs on FOC, determined by transmission electron microscopy (TEM).** CW, cell wall; cy, cytoplasm; pm, plasma membrane; M, mitochondrion. Bar: 0.5 μm. 2M4V: 2-Methoxy-4-vinylphenol; 34D: 3,4-Dimethoxystyrol; β-C: Caryophyllene.

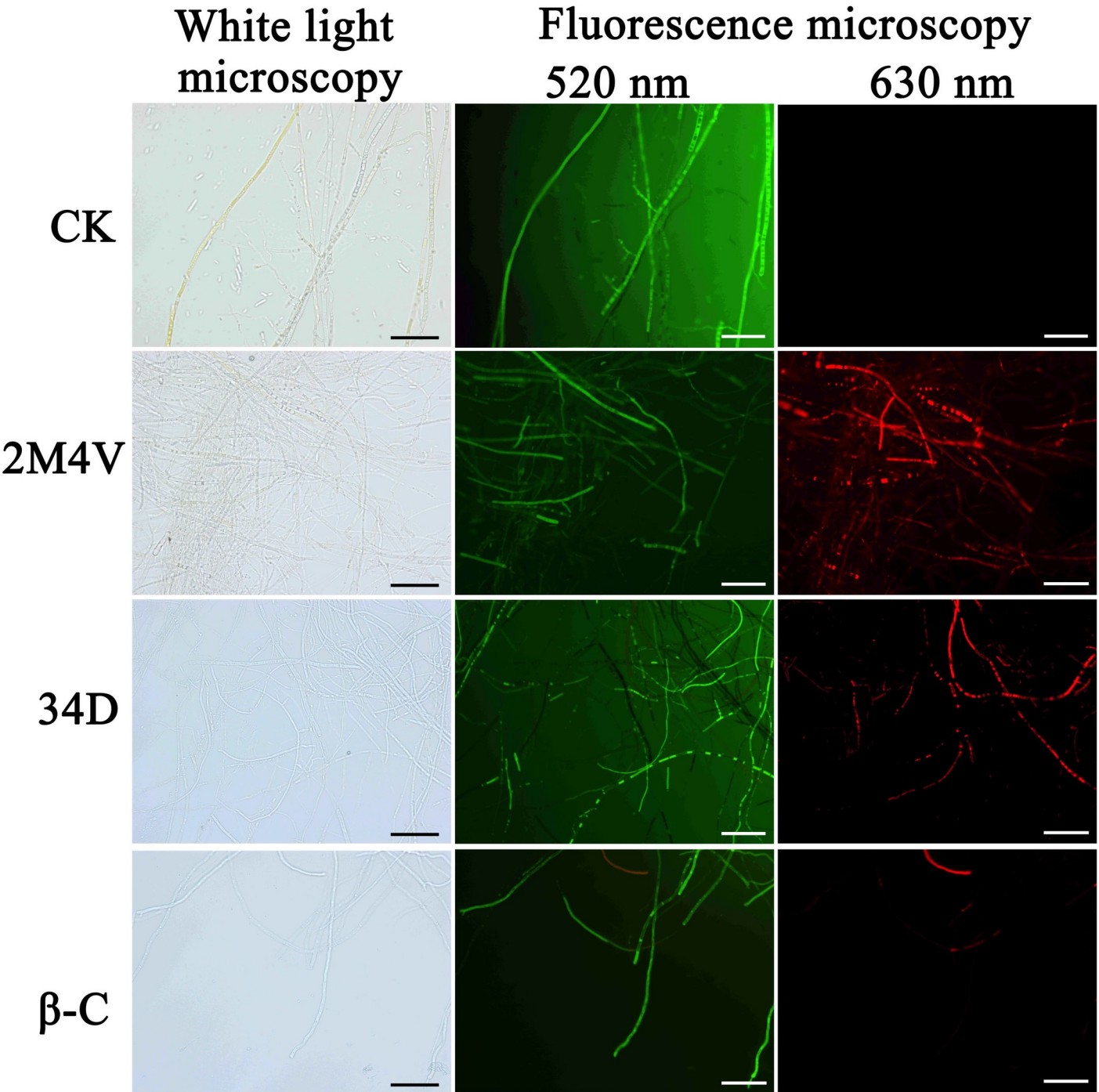

**Fig 5. Detection of FOC viability based on FDA/PI staining after treatment with selected VOCs.** Live fungal cells with intact membranes show green fluorescence; fungal cells with damaged membranes showed red fluorescence. Methanol served as the control (CK). Bar: 20 μm. 2M4V: 2-Methoxy-4-vinylphenol; 34D: 3,4-Dimethoxystyrol; β-C: Caryophyllene.

concentration, a DCFH-DA-based ROS assay kit was used. As shown in Fig 7, the untreated hyphae did not show any green fluorescence. Only a few cells treated with 2M4V or 34D showed green fluorescence. Unlike the other treatments, nearly all the β-C-treated mycelia

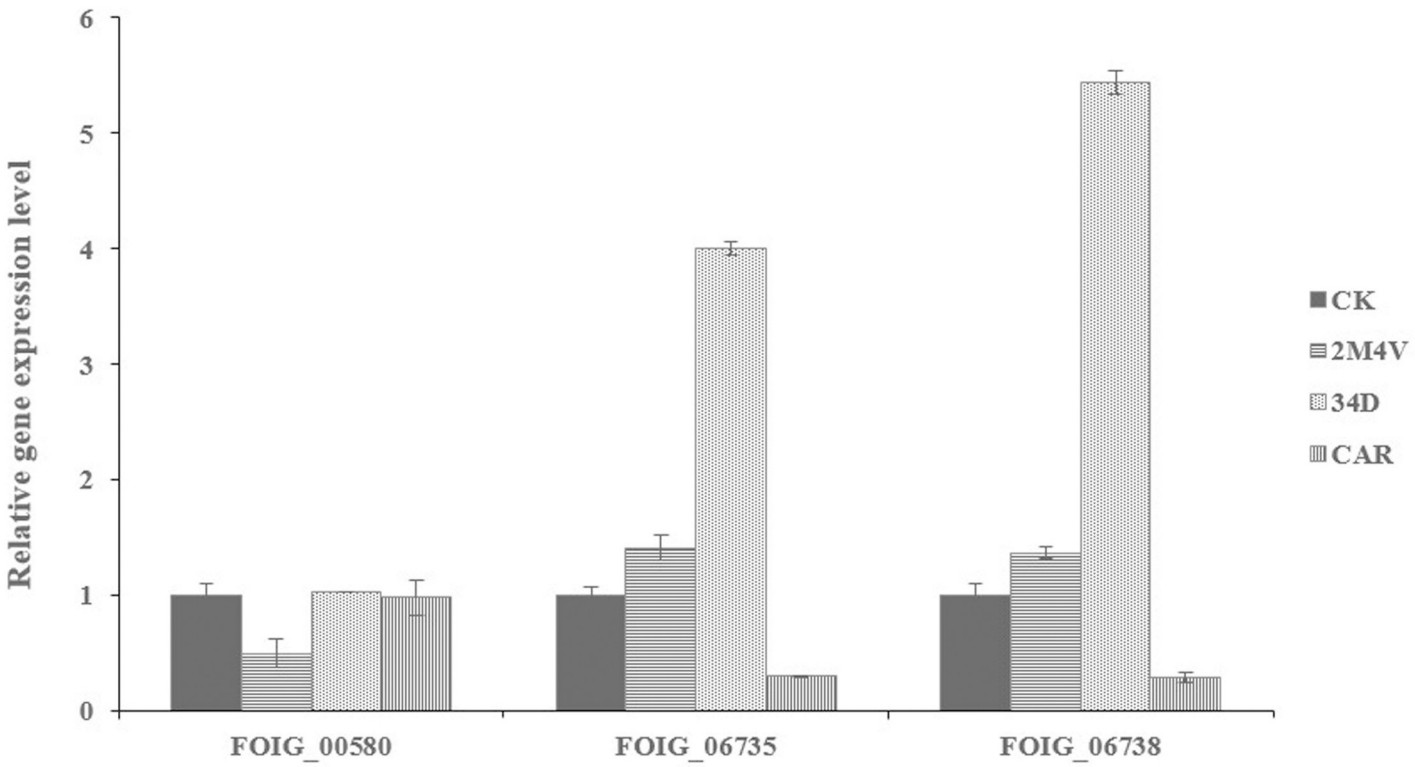

**Fig 6. Quantitative real-time PCR analysis of expression of three chitin synthesis genes (FOIG_00580, FOIG_06735 and FOIG_06738) in FOC in responsible to three selected VOCs.** Values were normalized to the levels of Actin as an internal reference gene. The y-axis represents the mean expression values ± SD relative to the control. The experiment was repeated independently three times. 2M4V: 2-Methoxy-4-vinylphenol; 34D: 3,4-Dimethoxystyrol; β-C: Caryophyllene.

showed strong green fluorescence. These findings indicate that treatment withβ-C can lead to the accumulation of ROS in FOC.

## Selected VOCs reduce fusaric acid production in FOC

To determine the involvement of the three selected VOCs in the biosynthesis of fusaric acid in FOC, we incubated FOC to a mix of sterilized wheat kernels/oats/corn (1:1:1) kernels with or without VOCs at the sublethal concentrations (2M4V and 34D: 1 μL/plate, β-C: 40 μL/plate) [39, 46]. After 23 days of incubation, the VOCs had significantly reduced the accumulation of fusaric acid in the kernels (Fig 8). In *Fusarium*, 1 gene cluster have been identified as the biosynthetic gene cluster of fusaric acid [47]. We analysed the expression levels of two genes of this cluster, FUB2 and FUB5. As shown in Fig 9, 2M4V, 34D, and β-C all deduced the expressions of FUB2 and FUB5. These results suggest that at sublethal concentrations (2M4V and 34D: 1 μL/plate, β-C: 40 μL/plate), the three selected VOCs can negatively affect fusaric acid biosynthesis.

## Discussion

FOC, which causes Fusarium wilt in banana, is the greatest threat to banana plantations worldwide [48, 49]. Low-toxicity and environmentally friendly biological control agents are required to control FOC. Because of their potential as biocontrol agents for fungal diseases, endophytic fungi have attracted considerable attention [49, 50]. *Sarocladium* spp. have been identified as a promising agents for the biocontrol of plant diseases [15, 51]. *Sarocladium brachiariae* HND5,

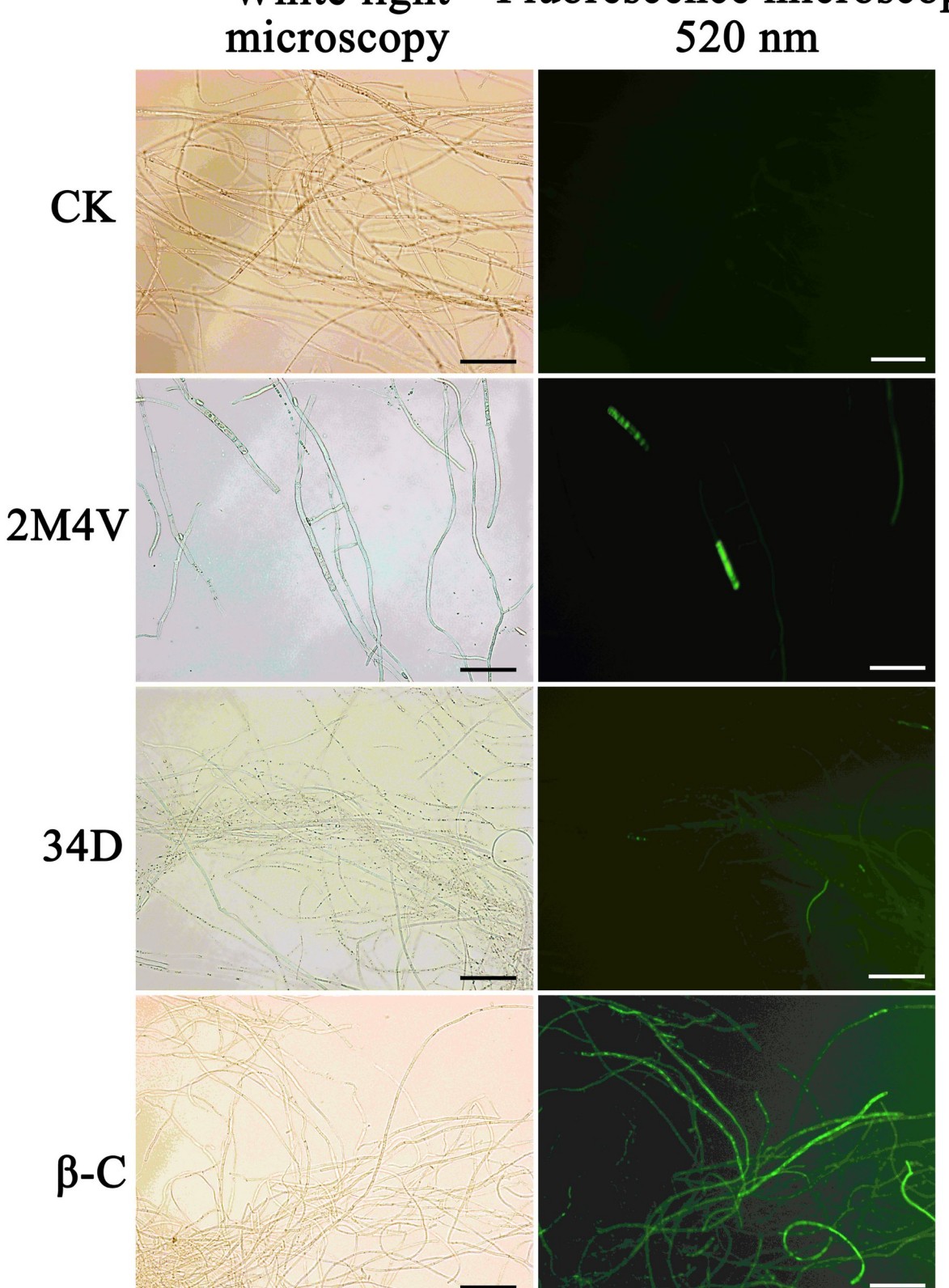

**Fig 7. Detection of ROS was based on DCFH-DA staining after treatment with selected VOCs for 12 h.** Methanol served as control (CK). Bar: 20 μm. 2M4V: 2-Methoxy-4-vinylphenol; 34D: 3,4-Dimethoxystyrol; β-C: Caryophyllene.

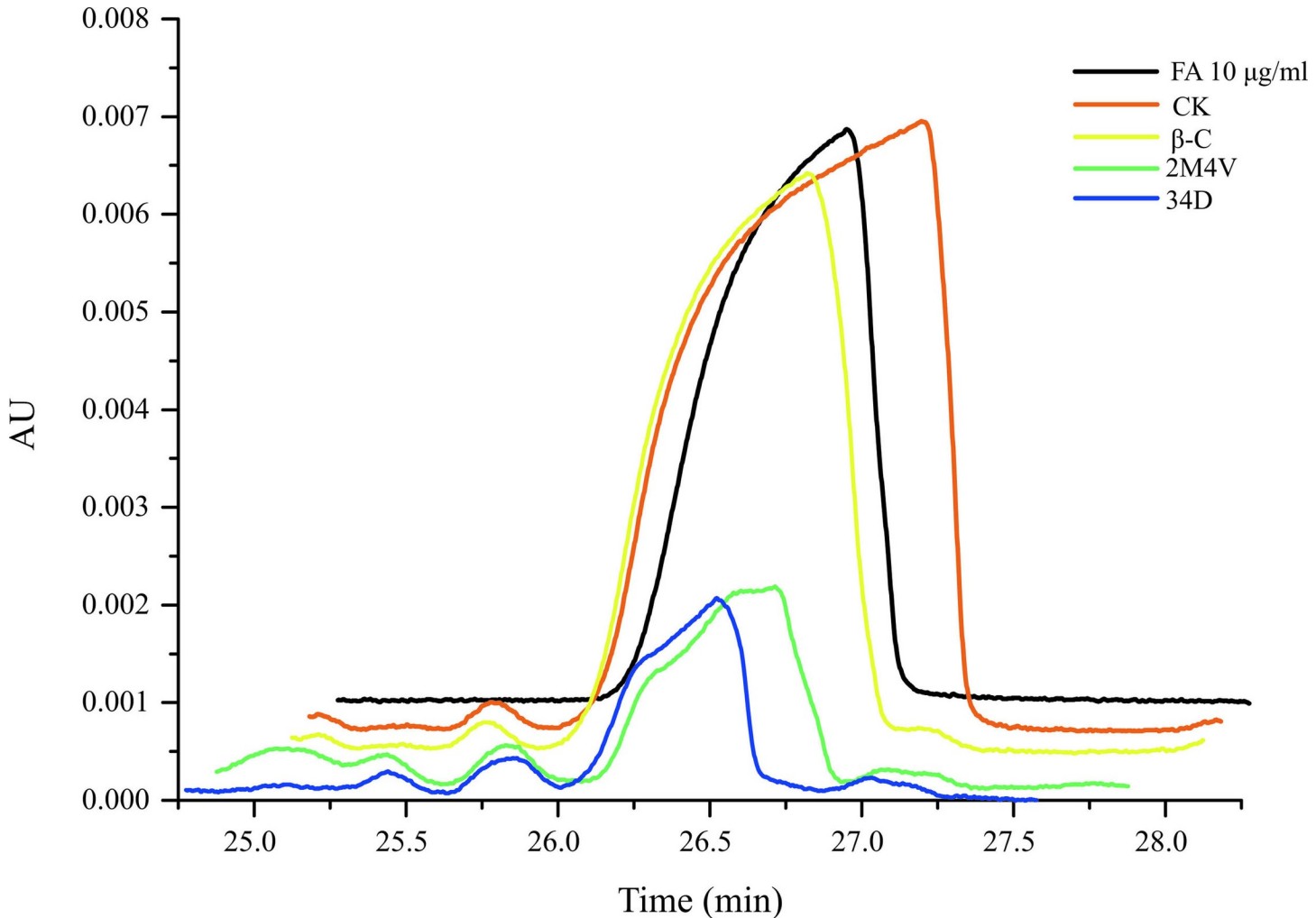

**Fig 8. Fusaric acid production was reduced by treatment with VOCs.** 2M4V: 2-methoxy-4-vinylphenol; 34D: 3,4-dimethoxystyrol; β-C: caryophyllene.

an endophytic fungus isolated from healthy *Brachiaria brizantha* leaf, is a new species of *Sarocladium* [28]. In this study, the VOCs emitted by HND5 effectively inhibited FOC growth. Using SPME-GC-MS, we identified 17 volatile compounds from the VOCs emitted by HND5. Among these 17 compounds, 2-methoxy-4-vinylphenol, 3,4-dimethoxystyrol, and caryophyllene were found to affect the growth of FOC. Further analysis indicated that these three VOCs produced cytoplasm leakage, ROS accumulation, and alterations to secondary metabolism. Identifying the active VOCs in HND5 and clarifying their mechanisms of action are critical for the application of such compounds in agriculture.

3,4-Dimethoxystyrol was the most abundant VOC (39% of total VOCs) emitted by HND5 and one of the most active compounds (EC50 = 60 μL/L headspace). A similar compound, 4-methoxystyrene, was recently identified in the VOCs produced by *Streptomyces albulus* NJZJSA2 [52]. 4-Methoxystyrene can inhibit growth and conidium germination in *Fusarium oxysporum* and *Sclerotinia sclerotiorum*, suggesting that this and similar compounds may have general antifungal activity. Although 2-methoxy-4-vinylphenol only accounted for 1.86% of total VOCs produced by HND5, it was the most active compound against FOC (EC50 = 36 μL/L headspace). 2-Methoxy-4-vinylphenol is typically found in wines as a flavour molecule

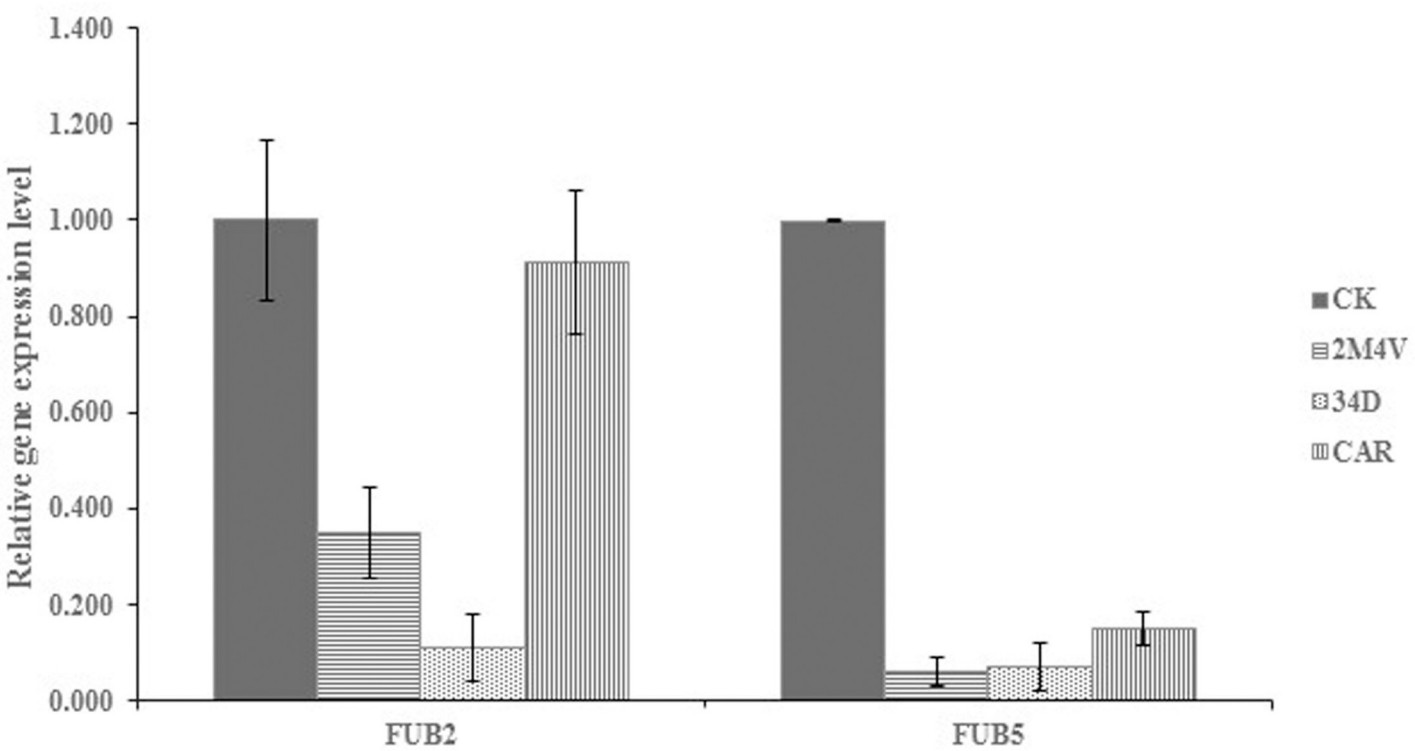

**Fig 9. Quantitative real-time PCR analysis of expression of two genes (FUB2 and FUB5) in FOC in responsible to three VOCs treatment.** Values were normalized to the levels of Actin as an internal reference gene. The y-axis represents the mean expression values ± SD relative to the control. The experiment was repeated independently three times. 2M4V: 2-Methoxy-4-vinylphenol; 34D: 3,4-Dimethoxystyrol; β-C: Caryophyllene.

and has been shown to have anti-inflammatory effects [53, 54]. Volatile sesquiterpenes have been identified in many fungi-produced VOCs, including *Streptomyces albulus*, *Fusarium oxysporum*, and *Gliocladium* sp. [52, 55, 56]. Two sesquiterpenes, caryophyllene and α-cadinol, were identified from HND5-generated VOCs. Based on anti-FOC assay, caryophyllene was found to have weak antifungal activity with EC50 > 2900 μL/L headspace, in agreement with a previous report [57]. We also identified three derivatives of naphthalene in the HND5-derived VOCs in this study. Naphthalene derivatives have been reported as an antimicrobial VOC in the essential oils of wood or volatile constituents of propolis [58].

TEM was used to study the ultra-structure of FOC after treatment with VOCs. 2M4V and 34D were found to induce cytoplasm leakage by disrupting the plasma membranes of FOC hyphae (Fig 4), consistent with the FDA/PI double fluorescence staining results (Fig 5). The plasma membrane is the target of many other antifungal VOCs, such as 4-methoxystyrene produced by *S. albulus* and oxygenated aromatic essential oil compounds [52, 59]. The TEM analyses also indicated the incrassation of FOC cell walls after treatment with 34D (Fig 4). Many other VOCs also affect cell walls in fungi, including the VOCs produced by *S. albulus* and farnesol. However, these compounds disrupt the integrity of the cell wall; they do not cause incrassation [60, 61]. Chitin is a major constituent of fungal cell walls synthesized by different types of chitin synthase genes. Besides of maintaining of cell wall integrity and structure, several chitin synthases play vital role in infection process of *Fusarium*, such as Class V, ChsVb and Class 4 types [42–44]. Class 4 chitin synthase co-regulate virulence, DON production and septum formation with chitin synthases in *F. graminearum* [45]. Class V and ChsVb chitin synthase is critical for pathogenicity and cell wall assembly in *F. oxysporum* [42, 43]. Marta et al. found Class V chitin synthase is hypersensitive to plant antimicrobial defence compounds

such as the tomato phytoanticipin a-tomatine or $H_2O_2$ and speculated that *F. oxysporum* requires a specific Class V chitin synthase for pathogenesis, most probably to protect itself against plant defence mechanisms [42]. Gene expression analysis indicated that expression levels of Class V (FOIG_06738), ChsVb (FOIG_06735) chitin synthase genes increased significantly after treatment with 34D at sublethal concentration (Fig 6), consistent with the TEM results (Fig 4). Based on these evidences, we conjecture that FOC identify 34D as plant antimicrobial defence compounds and activate defence system including cell wall enhancement. Low ROS concentrations act as intracellular messengers for many molecular events, whereas large amounts of ROS are associated with cell death [25, 62]. N-butanol, a volatile compound identified from *Muscodor albus*, induces ROC accumulation in bacteria [63]. Among the three VOCs evaluated in this study, only β-C caused the accumulation of ROS in FOC (Fig 7).

Fusaric acid is a well-known nonspecific toxin produced by all *Fusarium* species. Fusaric acid can kill banana cells and protoplasts and causes symptoms including the rotting of roots and pseudostems and the wilting of seedling leaves. Siwen Liu et al. found both banana leaves and pseudostems exhibited increased sensitivity to Foc4 invasion when pretreated with fusaric acid and suggested that fusaric acid functions as a positive virulence factor and acts at the early stage of the disease development before the appearance of the fungal hyphae in the infected tissues [46]. Although fusaric acid is not considered to be a mycotoxin with significant health consequences to humans, it still causes pathological disorders in experimental animals and human cell lines [64]. In this study, we found that treatment with 2M4V, 34D and β-C decreased the production of fusaric acid in FOC (Fig 8). Previous studies revealed that the production of fusaric acid is encoded by the fusaric acid biosynthetic gene cluster containing 12 genes (FUB1-FUB12) [47, 65]. But the biosynthesis progress of fusaric acid is not clear yet. Yang Hai et al. elucidated the biosynthesis of fusaric acid by reconstitution of the biosynthesis gene cluster in *Aspergillus nidulans* and precursor feeding. Based on their result, 8 genes (FUB1, FUB3-9) of fusaric acid biosynthetic gene cluster are responsible for the synthesis [47]. Siwen Liu et al. found FUB1-5 and FUB10 involved in the biosynthesis of fusaric acid by construction of deletion mutants [47]. On the basis of the above research, we chose to analyse the expression of FUB2 and FUB5 to evaluate the effect of target VOCs on fusaric acid biosynthesis. Gene expression assay demonstrated that 2M4V, 34D and β-C all decreased the expressions of FUB2 and FUB5 (Fig 9) at sublethal concentration, consistent with the HPLC results (Fig 8). These results indicated that target VOCs could influence the biosynthesis of fusaric acid and this indicated that *S. brachiariae* HND5 could delay the invasion of banana by FOC by decreasing the production of fusaric acid with VOCs.

As FOC is a soil-born pathogen, soil sterilizers and fumigants are frequently used to control this pathogen [66]. Methyl bromide is an effective fumigant against soil-borne pathogens and was broadly used worldwide on many crops until 2015, when it was phased out because it depletes the ozone layer [67, 68]. Many chemicals have been studied as alternatives to methyl bromide, including metham sodium, 1,3-dichloropropene, chloropicrin, sulfuryl fluoride and methyl iodide [69, 70]. In this study, three antifungal VOCs were identified from HND5. Of these, two possess high anti-FOC activity and show potential as methyl bromide alternatives. The mechanisms of antifungal activity of these VOCs against FOC were also clarified. The findings suggest that HND5 and the VOCs it generates show promise for use as biological control agents or fumigants against FOC in agricultural production systems.

## Conclusion

This study identified seventeen compounds from the volatile organic compounds (VOCs) produced by endophytic fungi *Sarocladium brachiariae* HND5. Three VOCs of the seventeen

(2-methoxy-4-vinylphenol, 3,4-dimethoxystyrol and caryophyllene) showed antifungal activity against *Fusarium oxysporum* f. sp. *cubense* (FOC)with 50% effective concentrations of 36, 60 and 2900 µL/L headspace, respectively. Transmission electron microscopy (TEM) and double fluorescence staining revealed that 2-methoxy-4-vinylphenol and 3,4-dimethoxystyrol damaged the plasma membranes, resulting in cell death. 3,4-dimethoxystyrol also could induce expression of chitin synthases genes and altered the cell walls of FOC hyphae. Dichloro-dihydro-fluorescein diacetate staining indicated the caryophyllene-induced accumulation of reactive oxygen species (ROS) in FOC hyphae. All three target VOCs could decrease biosynthesis of fusaric acid at sublethal concentration.

## Supporting information

**S1 Table. Gas chromatography/mass spectrometry (GC/MS) analysis result of HND5 culture and PDA medium (SPME fibre:PDMS, 100 µm).**
(XLS)

**S2 Table. Gas chromatography/mass spectrometry (GC/MS) analysis result of HND5 culture and PDA medium (SPME fibre: DVB, 65 µm).**
(XLS)

**S3 Table. Gas chromatography/mass spectrometry (GC/MS) analysis result of HND5 culture and PDA medium (SPME fibre: DVB/CAR/PDMS, 50/30 µm).**
(XLS)

**S4 Table. Primers used in this study.**
(DOCX)

## Author Contributions

**Data curation:** Yang Yang.

**Funding acquisition:** Yang Yang.

**Investigation:** Yipeng Chen.

**Resources:** Jimiao Cai.

**Software:** Yipeng Chen.

**Supervision:** Guixiu Huang.

**Validation:** Xianbao Liu.

**Visualization:** Jimiao Cai.

**Writing – original draft:** Yang Yang.

**Writing – review & editing:** Xianbao Liu.

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
