## [Decision Letter · Decision Letter 0]

28 Sep 2021

PONE-D-21-24535Antifungal activity of volatile compounds generated by endophytic fungi HND5 against Fusarium oxysporum f. sp. cubensePLOS ONE

Dear Dr. Huang,

Thank you for submitting your manuscript to PLOS ONE. After careful consideration, we feel that it has merit but does not fully meet PLOS ONE’s publication criteria as it currently stands. Therefore, we invite you to submit a revised version of the manuscript that addresses the points raised during the review process.

We look forward to receiving your revised manuscript.

Kind regards,

Vivek Sharma, PhD

Academic Editor

PLOS ONE

Journal Requirements:

"This study was supported in part by grants from Hainan Provincial Natural Science Foundation of China, 319QN268"

"YY is founded by Hainan Provincial Natural Science Foundation of China [Grant 319QN268]. The funders had no role in study design, data collection and analysis, decision to publish, or preparation of the manuscript"

Additional Editor Comments (if provided):

Dear Authors,

I am pleased to inform that your MS ID PONE-D-21-24535 entitled "Antifungal activity of volatile compounds generated by endophytic fungi HND5 against Fusarium oxysporum f. sp. cubense" can be accepted for publication after after point-wise corrections to the reviewers suggestions. Both the reviewers have suggested minor revision.

Reviewers' comments:

Reviewer's Responses to Questions

**Comments to the Author**

1. Is the manuscript technically sound, and do the data support the conclusions?

Reviewer #1: Yes

Reviewer #2: Yes

2. Has the statistical analysis been performed appropriately and rigorously? 

Reviewer #1: Yes

Reviewer #2: N/A

3. Have the authors made all data underlying the findings in their manuscript fully available?

Reviewer #1: Yes

Reviewer #2: Yes

4. Is the manuscript presented in an intelligible fashion and written in standard English?

Reviewer #1: No

Reviewer #2: Yes

5. Review Comments to the Author

Reviewer #1: The authors have presented here the possibility of employing the volatile organic compounds from endophytic fungus Sarocladium brachiariae HND5 to antagonise the pathogen Fusarium oxysporum f. sp. Cubense of banana plants. The proper experimental approach has been carried out to support the findings.

There are few suggestions:

The introduction of the manuscript can be improved and made more informative for the reader. Introduction should reflect the extent of work that is carried out. Such as there is no mention of Fusaric acid and its role in disease development in the introduction portion to reflect the need to study its production.

The name of the fungus Sarocladium brachiariae has to be italicised in the manuscript. Also scientific names in some other places has to be italicised.

The following lines from Introduction portion (pg 7 and 8) are repeated:

This pathogen can infect banana from root and invade the xylem vessels and eventually cause a lethal wilting of the infected plants [3]. This pathogen can infect banana plants from the root and invade the xylem vessels, eventually causing lethal wilting of the infected plant2.

Such repetitions of statements are in end of Discussion and Conclusion section too (pg 20 and 21): ‘VOCs it generates show promise for use as biological control agents or fumigants against FOC in agricultural production systems.’ It should be avoided.

The line in the Material and Methods is a repeat from introduction: The antagonistic strain HND5, which was isolated from healthy leaf of Brachiaria brizantha, was identified as Sarocladium brachiariae. (China General Microbiology Culture Collection Center accession No. 2192) according to the LSU and ITS rDNA sequence [30, 31].

It needs to be restructured according to the Material and methods section.

There are certain grammatical mistakes as hyphen has been used in between some of the words such as effec-tive, ac-id etc. in the Abstract.

Reviewer #2: A quality of work has been done by authors and well written manuscript except few mistakes done.

Firstly, add name of fungi in the title of manuscript instead of only writing HND5,

Remove hyphen wherever not required as yellow highlighted,

Write fungi name in italics,

Remove repeated sentance,

Write incubated/ion instead of inoculated/ion wherever required,

Write "parafilm/ petri" in small letters,

Write PBS, 2M4V, 34D, B-C in bracket when first appear in text in full form,

There are some minor speeling errors like regent, unite which should be corrected to reagent, unit,

Its better to use "C" for control instead of "CK",

6. PLOS authors have the option to publish the peer review history of their article (what does this mean?). If published, this will include your full peer review and any attached files.

Reviewer #1: **Yes: **Banita Kumari Saklani

Reviewer #2: **Yes: **Dr, Zalak M Patel

---

## [Author Response · Author response to Decision Letter 0]

9 Oct 2021

Dear Editor,

Thank you for serving as the editor of our manuscript and managing the review process. We also acknowledge the efforts of the reviewers that you assembled and appreciate their constructive comments and suggestions to improve the quality of our manuscript. We have made the suggested corrections, which are provided in the revised manuscript and specific responses to each of the reviewer’s comments can be found below. Thank you for your further consideration of our revised manuscript. 

Detail response:

Reviewer 1: The authors have presented here the possibility of employing the volatile organic compounds from endophytic fungus Sarocladium brachiariae HND5 to antagonise the pathogen Fusarium oxysporum f. sp. Cubense of banana plants. The proper experimental approach has been carried out to support the findings.

There are few suggestions:

The introduction of the manuscript can be improved and made more informative for the reader. Introduction should reflect the extent of work that is carried out. Such as there is no mention of Fusaric acid and its role in disease development in the introduction portion to reflect the need to study its production.

Answer: Thank you for your constructive criticism and detailed review of this manuscript. We have added description of fusaric acid in introduction portion.

The name of the fungus Sarocladium brachiariae has to be italicised in the manuscript. Also scientific names in some other places has to be italicised.

Answer: We have checked all thquanrough the manuscript and changed all scientific names to italic.

The following lines from Introduction portion (pg 7 and 8) are repeated:

This pathogen can infect banana from root and invade the xylem vessels and eventually cause a lethal wilting of the infected plants [3]. This pathogen can infect banana plants from the root and invade the xylem vessels, eventually causing lethal wilting of the infected plant2.

Such repetitions of statements are in end of Discussion and Conclusion section too (pg 20 and 21): ‘VOCs it generates show promise for use as biological control agents or fumigants against FOC in agricultural production systems.’ It should be avoided.

Answer: We have deleted the repeated sentences.

The line in the Material and Methods is a repeat from introduction: The antagonistic strain HND5, which was isolated from healthy leaf of Brachiaria brizantha, was identified as Sarocladium brachiariae. (China General Microbiology Culture Collection Center accession No. 2192) according to the LSU and ITS rDNA sequence [30, 31].

It needs to be restructured according to the Material and methods section.

Answer: This sentence has been restructured according to the Material and methods section.

There are certain grammatical mistakes as hyphen has been used in between some of the words such as effec-tive, ac-id etc. in the Abstract.

Answer: We have checked all through the manuscript and deleted all misused hyphens. 

Reviewer 2: A quality of work has been done by authors and well written manuscript except few mistakes done.

Firstly, add name of fungi in the title of manuscript instead of only writing HND5,

Answer: Thank you for your comments. They are very helpful for improving our manuscript. The fungi name has been added in the title of manuscript.

Remove hyphen wherever not required as yellow highlighted,

Answer: We have checked all through the manuscript and deleted all misused hyphens.

Write fungi name in italics,

Answer: We have checked all through the manuscript and changed all scientific names to italic.

Remove repeated sentance,

Answer: We have removed repeated sentences in the introduction part.

Write incubated/ion instead of inoculated/ion wherever required,

Answer: We have changed inoculated into incubated.

Write "parafilm/ petri" in small letters,

Answer: We have checked all through the manuscript and changed "parafilm/ petri" into small letters.

Write PBS, 2M4V, 34D, B-C in bracket when first appear in text in full form,

Answer: We have added full name of PBS, 2M4V, 34D, B-C in the text where first apper.

There are some minor speeling errors like regent, unite which should be corrected to reagent, unit,

Answer: We have checked all through the manuscript and corrected these spelling mistakes.

Its better to use "C" for control instead of "CK"

Answer: As many academic papers using “CK” for control, we keep “CK” in revised manuscript. Thanks for your suggestion!

---

## [Editor Report · Decision Letter 1]

26 Oct 2021

PONE-D-21-24535R1Antifungal activity of volatile compounds generated by endophytic fungi Sarocladium brachiariae HND5 against Fusarium oxysporum f. sp. cubensePLOS ONE

Dear Dr. Huang,

Thank you for submitting your manuscript to PLOS ONE. After careful consideration, we feel that it has merit but does not fully meet PLOS ONE’s publication criteria as it currently stands. Therefore, we invite you to submit a revised version of the manuscript that addresses the points raised during the review process.

We look forward to receiving your revised manuscript.

Kind regards,

Vivek Sharma, PhD

Academic Editor

PLOS ONE

Journal Requirements:

Additional Editor Comments (if provided):

Dear Authors,

I am happy to share that both the reviewers have recommended your publication with minor revision. The details comments can be found in the reviewers section.
---

## [Author Response · Author response to Decision Letter 1]

8 Nov 2021

Dear Editor,

Thank you for serving as the editor of our manuscript and managing the review process. We also acknowledge the efforts of the reviewers that you assembled and appreciate their constructive comments and suggestions to improve the quality of our manuscript. We have reviewed reference list and did not find retracted papers. Also, we have used PACE to adjust figures and uploaded adjusted figures in the revised manuscript. Thank you for your further consideration of our revised manuscript. 

Detail response:

Reviewer 2: 

Title (Page 1): Full name of fungi (Sarocladium brachiariae) have been inserted.

Abstract (Page 1): Misused hyphens have been removed. And fungi name have been written in italics.

Introduction (Page 2): We have removed repeated sentences in the introduction part.

Introduction (Page 2): We have checked and make sure is “low target specificity”.

Materials and Methods (Page 4): We have changed "parafilm/ petri" into small letters and corrected “inoculate” to “incubate”.

Materials and Methods (Page 5): We have changed "parafilm/ petri" into small letters and corrected “inoculate” to “incubate”. And we also written PBS in text in full form.

Materials and Methods (Page 6): We have changed "parafilm/ petri" into small letters and corrected “inoculate” to “incubate”. And we also written 2M4V, 34D, B-C in text in full form.

Results (Page 7): We have changed "parafilm/ petri" into small letters, corrected “inoculate” to “incubate” and removed misused hyphen.

Results (Page 8): We have changed "regent" into “reagent”.

Results (Page 9): We have changed "unite" into “unit” and corrected “inoculate” to “incubate”.

Results (Page 11): We have removed misused hyphen.

Figures: As many academic papers using “CK” for control, we keep “CK” in revised manuscript. Thanks for your suggestion!

---

## [Editor Report · Decision Letter 2]

17 Nov 2021

Antifungal activity of volatile compounds generated by endophytic fungi Sarocladium brachiariae  HND5 against Fusarium oxysporum f. sp. cubense

PONE-D-21-24535R2

Dear Dr. Huang,

We’re pleased to inform you that your manuscript has been judged scientifically suitable for publication and will be formally accepted for publication once it meets all outstanding technical requirements.

For more information, please contact onepress@plos.org.

Kind regards,

Vivek Sharma, PhD

Academic Editor

PLOS ONE

Additional Editor Comments (optional):

Dear Dr. Huang,

I am please to inform that your MS has been accepted for publication in PLOS One.
---

## [Editor Report · Acceptance letter]

24 Nov 2021

PONE-D-21-24535R2 

Antifungal activity of volatile compounds generated by endophytic fungi *Sarocladium brachiariae* HND5 against *Fusarium oxysporum* f. sp. *cubense*

Dear Dr. Huang:

I'm pleased to inform you that your manuscript has been deemed suitable for publication in PLOS ONE. Congratulations! Your manuscript is now with our production department. 

Kind regards, 

on behalf of

Dr. Vivek Sharma 

Academic Editor

PLOS ONE